# MicroRNA Let-7a, -7e and -133a Attenuate Hypoxia-Induced Atrial Fibrosis via Targeting Collagen Expression and the JNK Pathway in HL1 Cardiomyocytes

**DOI:** 10.3390/ijms23179636

**Published:** 2022-08-25

**Authors:** Chien-Hsien Lo, Li-Ching Li, Shun-Fa Yang, Chin-Feng Tsai, Yao-Tsung Chuang, Hsiao-Ju Chu, Kwo-Chang Ueng

**Affiliations:** 1Institute of Medicine, Chung Shan Medical University, Taichung 402, Taiwan; 2Division of Cardiology, Department of Internal Medicine, Chung Shan Medical University Hospital, Taichung 402, Taiwan; 3Division of Endocrinology, Department of Internal Medicine, Chung Shan Medical University Hospital, Taichung 402, Taiwan; 4Department of Medical Research, Chung Shan Medical University Hospital, Taichung 402, Taiwan; 5School of Medicine, Chung Shan Medical University, Taichung 402, Taiwan

**Keywords:** microRNA, JNK pathway, atrial fibrosis

## Abstract

Fibrosis is a hallmark of atrial structural remodeling. The main aim of this study was to investigate the role of micro-ribonucleic acids (miRNAs) in the modulation of fibrotic molecular mechanisms in response to hypoxic conditions, which may mediate atrial fibrosis. Under a condition of hypoxia induced by a hypoxia chamber, miRNA arrays were used to identify the specific miRNAs associated with the modulation of fibrotic genes. Luciferase assay, real-time polymerase chain reaction, immunofluorescence and Western blotting were used to investigate the effects of miRNAs on the expressions of the fibrotic markers collagen I and III (COL1A, COL3A) and phosphorylation levels of the stress kinase c-Jun N-terminal kinase (JNK) pathway in a cultured HL-1 atrial cardiomyocytes cell line. COL1A and COL3A were found to be the direct regulatory targets of miR-let-7a, miR-let-7e and miR-133a in hypoxic atrial cardiac cells in vitro. The expressions of COL1A and COL3A were influenced by treatment with miRNA mimic and antagomir while hypoxia-induced collagen expression was inhibited by the delivery of miR-133a, miR-let-7a or miR-let-7e. The JNK pathway was critical in the pathogenesis of atrial fibrosis. The JNK inhibitor SP600125 increased miRNA expressions and repressed the fibrotic markers COL1A and COL3A. In conclusion, MiRNA let-7a, miR-let-7e and miR-133a play important roles in hypoxia-related atrial fibrosis by inhibiting collagen expression and post-transcriptional repression by the JNK pathway. These novel findings may lead to the development of new therapeutic strategies.

## 1. Introduction

Atrial fibrillation (AF) is becoming an enormous public-health challenge and a major contributor to cardiovascular morbidity, such as stroke, dementia, heart failure and mortality [1]. Deficient oxygen supply may promote atrial structural remodeling, fibrogenesis and myocardial dysfunction [2,3], leading to the development of AF, especially in selective cardiac atrial ischemia. Higher expressions of hypoxic and angiogenic markers have been reported in patients with AF and up-regulation of the hypoxia-inducible factor (HIF) pathway with fibrogenesis in atrial myocardium have also been reported [3,4]. There is a need for a better understanding of the causes and consequences of the hypoxia-related development of AF and atrial remodeling (i.e., fibrosis). However, the pathogenetic mechanisms underlying atrial structural remodeling remain poorly understood. Atrial structural remodeling is characterized by increased interstitial fibrosis, extracellular matrix (ECM) deposition and hypertrophy. Abundant deposition of the ECM has been shown in the maturation of fibrosis and the ratio of collagen type I has been shown to be increased in fibrosis [5]. In our previous research, we concluded that aldosterone significantly increased the protein expressions of collagen I, III (COL1A, COL3A), transforming growth factor (TGF)-β1 and α-smooth muscle actin (SMA), which demonstrated the critical role of mineralocorticoid receptor activity in aldosterone-mediated activation of the mitogen-activated protein kinase (MAPK) signaling pathway and subsequent atrial profibrotic effects [6]. Furthermore, it was shown that hypoxia resulted in cardiac fibrosis by inducing the up-regulation of COL1A and COL3A in atrial cardiac muscle (HL-1) cells.

Micro-ribonucleic acids (miRNAs) have been reported to have an impact on the pathogenesis of cardiac diseases [7,8] and recent evidence has indicated that miRNAs participate in the development and progression of cardiovascular diseases [7,8,9,10,11,12,13,14]. However, miRNAs should be fully exploited—how their targets functionally interact with and are simultaneously regulated by multiple miRNAs. Furthermore, few studies have explored the role of miRNAs in fibrosis in response to hypoxic stress in cardiomyocytes. Therefore, we focused on miRNAs in this study and used microarray analysis to identify the differential expressions of several miRNAs in hypoxic cardiomyocytes. Consequently, we hypothesized that some specific miRNAs may play a significant role in regulating fibrosis and cardiac remodeling under hypoxic stress.

Evidence that specific miRNAs can influence AF related to atrial structural remodeling and fibrosis via the modulation of collagens under myocardial hypoxic conditions is lacking. Therefore, the aim of this study was to investigate two major hypotheses: (i) whether miRNAs modulate fibrotic molecular mechanism responses to hypoxic conditions and whether this mediates cardiac remodeling in cardiomyocytes; and (ii) whether a miRNA antagomir or miRNA mimic acted on target genes under hypoxic conditions in cardiomyocytes.

## 2. Results

### 2.1. Fibrotic Markers Were Up-Regulated in Hypoxic HL-1 Cells

Western blot analysis and real-time RT-PCR were used to explore the role of fibrotic markers in cardiomyocyte fibrosis induced by hypoxia. As shown in Figure 1A,B, HIF-1α, COL1A and COL3A expressions were up-regulated by hypoxia in a time-dependent manner.

### 2.2. MiR-Let-7 Family Expression in Hypoxic Cardiomyocytes

To clarify the hypoxic effects of regulating myocardial fibrosis by miRNA, miRNA microarray analysis was performed to reveal differential expressions of several miRNAs in hypoxic cardiomyocytes, including miR-5099, miR-let-7e-5p, miR-5117-3p, miR-711, miR-669d, miR-5129, miR-2137, miR-3960, miR-1949, miR-883a-3p and miR-1983 (Figure 1C). Appendix A lists the different miRNA expressions of log2 ≤ 0.5 and ≥0.3 under hypoxia/control detected by miRNA microarray assay. MiR-let-7a and miR-let-7e were found to be expressed as log2 0.53 and 0.69, respectively.

### 2.3. Transcriptional Effects of MiR-133a, MiR-let-7a and MiR-Let-7e Clusters in Hypoxic State

Considering that cardiomyocytes express different fibrosis markers, COL1A and COL3A expression under low-oxygen-stress conditions, we compared the miRNA microarray profile of each marker with normoxic cardiomyocytes to evaluate whether the expression profiles of miRNA targeting COL1A and COL3A were different from those in the hypoxic-stressed cardiomyocytes. Identification of the miRNA targeting COL1A and COL3A was conducted by searching miRDB and TargetScan85 and proteins regulated by one or multiple miRNAs in our miRNA microarray list had higher target scores in either or both searches (Figure 1C). The miRNAs with low expressions in hypoxic cells are shown in the Venn diagram in Figure 1D, in which 2 of 19 miRNAs targeting COL1A and COL3A were significantly down-modulated in a hypoxic state compared to that in normoxic cardiomyocytes. The Venn diagram of miRNA expression profiles indicated two common miRNAs (miR-let-7a and let-7e), with 39 and 64 unique miRNAs targeting COL1A and COL3A, respectively. The three circles generated by TargetScan (http://targetscan.org) identified that miR-let-7 targeted COL1A and COL3A in both the online miRNA databases and microarray data. We also found that both miR133a and miR-let-7 (including let-7a and let-7e) were significantly suppressed by hypoxic stress, as demonstrated by qRT-PCR (Figure 1E). Based on these findings, we hypothesize that fibrotic markers (i.e., COL1A and COL3A) are associated with miR-let-7a and miR-let-7e target genes.

### 2.4. Post-Transcriptional Repression of Collagen Type 1A1 and Collagen Type 3A1 by MiR-Let-7a, MiR-Let-7e and MiR-133a

To further elucidate whether miR-133a and miR-let-7 interfere with COL1A1 and COL3A1 3′-UTR, pGL4.13-based plasmids containing binding sites for miR-133a or miRlet-7 were constructed. The 3′-UTR of COL3A1 also had putative binding sequences for miR-let-7 (Figure 2a), while COL1A1 and COL3A1 genes contained putative binding sites for miR-133a in the 3′-UTR region (Figure 2c). Subsequently, the ability of miR-133a and miR-let-7 to repress COL1A1 or COL3A1 expression was evaluated. In Figure 2b, a luciferase vector carrying the 3′-UTR of COL1A or COL3A was co-transfected with miR-let-7 mimic and inhibitor. Results indicated that after a co-transfection with miR-let-7 mimic, the luciferase vector carrying the 3′-UTR of COL1A or COL3A was co-transfected and luciferase activity was significantly diminished, compared to that after transfection with a negative control mimic. The inhibitory effect of miR-let-7 on COL1A and COL3A was antagonized by the miRNA inhibitor to induce COL1A or COL3A 3′-UTR luciferase expression. Similar results were found when COL1A or COL3A was co-transfected with miR-133a mimic and inhibitor (Figure 2d). There was also no significant difference in luciferase activity between plasmids containing a mutation of the COL1A or COL3A 3′-UTR + miRNA negative control (COL1A or COL3A let7-mut + NC) and the plasmid containing a mutation of the COL1A 3′-UTR + miRNA mimic or inhibitor (COL1A or COL3A let7-mut + miR-let-7 mimic, COL1A or COL3A let7-mut + miR-let-7 inhibitor in Figure 2b), (COL1A or COL 3A 133a-mut + miR-133a mimic, COL1A or COL 3A 133a-mut + miR-133a inhibitor in Figure 2d). All of these results confirmed that COL1A and COL3A were the targets of miR-133a and miR-let-7.

### 2.5. Delivery of MiR-133a, MiR-Let-7a or MiR-Let-7e Inhibited Hypoxia-Induced Collagen Expression

HL-1 cells transfected with miR-133a, miR-let-7a or miR-let-7e were exposed to a hypoxic microenvironment and then analyzed using Western blot and qRT-PCR. Results in Figure 3a shows that endogenous miR-133a, miR-let-7a and miR-let-7e expressions in HL-1 cells were up-regulated after transfection with an miR-133a, miR-let-7a and miR-let-7e mimic. Moreover, after a transfection with miR-133a, miR-let-7a or miR-let-7e mimic (100 nM), the expressions of proCOL1A and proCOL3A were repressed (Figure 3b,c). Furthermore, results of qRT-PCR demonstrated decreased transcription levels of fibrotic markers by miRNA (miR-133a, miR-let-7a, miR-let-7e) mimic modulation under hypoxic stress (Figure 3d). Results in Figure 3e demonstrates that the hypoxia-induced expressions of COL1A and COL3A were decreased by a co-transfection with miR-133a, miR-let-7a or miR-let-7e mimic in hypoxia HL-1 cells. As mentioned above, COL1A and COL3A are targets of miR-133A, miR-let-7a and miR-let-7e and, therefore, these data strongly support the anti-fibrotic role of miR-133a, miR-let-7a and miR-let-7e under hypoxic stress.

### 2.6. Post-Transcriptional Repression of MiRNAs by the JNK Pathway

Results in Appendix A show the most important pathways related to miRNA expression, which includes the MAPK signaling pathway. Since the MAPK pathway has been implicated to play an important role in the pathogenesis of cardiac diseases, we used Western blot analysis to further investigate the underlying molecular mechanisms. As shown in Figure 4a,b, the phosphorylation levels of the stress kinase JNK pathway (JNK, ATF2 and MKK4) were significantly higher under hypoxic conditions than those of the controls. Furthermore, to investigate whether the activation of JNK phosphorylation by hypoxia interferes with the actions of miRNA and fibrotic markers in HL-1 cells, an inhibitor of JNK1/2 (SP600125) was employed. As shown in Figure 4c,d, miR-let-7a, miR-let-7e, miR-133a, COL1A and COL3A were activated under hypoxic conditions. Intriguingly, JNK1/2 inhibitor (SP600125) significantly repressed the hypoxia-related increase in miRNA and fibrotic markers in the HL-1 cells. Overall, these findings indicated that JNK1/2 pathways play a critical upstream role in hypoxia-mediated atrial fibrosis in HL-1 cells. There was a link between the JNK pathway and miRNA expression as well as the changes in fibrotic marker level.

## 3. Discussion

Cardiac hypoxia can promote cardiac remodeling, the development of heart failure and dysrhythmias, such as AF [2,3,15]. Strong evidence indicates that the pronounced simultaneous expression of both COL1A and COL3A is an important contributor to the AF substrate [16]. However, the mechanisms of the development of atrial fibrosis are still not completely understood. In this study, we identified a linkage between miRNAs and cardiac fibrosis and evaluated the role of miRNA in the modulation of ECM protein (COL1A and COL3A) in hypoxia-treated cardiomyocytes (HL-1 cells).

Several recent studies have supported that some miRNAs play important roles in cardiac diseases, including cardiac remodeling, arrhythmia, cardiac hypertrophy and heart failure [11,17,18,19,20,21,22]. For example, miR-208 is cardiac specific and has been implicated to play a role in cardiac hypertrophy and fibrosis [17]. In animal experiments of cardiac hypertrophy and heart failure, miR-21, miR-23a, miR-125 and miR-27 were related with profibrotic roles, while miR-1, miR-29, miR-30, miR-150 and miRNA-133 have been reported to play anti-fibrotic roles [14,23,24,25]. In addition, miR-21 and miR-24 have been shown to be up-regulated in myocardial infarction (MI) [26,27] and miR-29 has been shown to be down-regulated after MI and to play a role in cardiac fibrosis by negatively regulating mRNAs encoding various collagens [28,29,30]. In an animal model and human atrial samples from AF patients, the expressions of miR-328, miR-223 and miR-664 were found to be higher, whereas miR-101, miR-320 and miR-499 were down-regulated by at least 50% [31]. Moreover, the overexpression of miR-133 has been shown to attenuate cardiac hypertrophy [32] and miR-133A has been shown to target the 3′-UTR of COL1A and to be down-regulated in the presence of myocardial fibrosis [23,33]. Wang et al. reported that miR-155, miR-146b and miR-19b were significantly up-regulated in non-valvular AF patients [34]. In addition, Nicoline et al. reviewed the role of various miRNAs, including miR-328, miR-208, miR-21, miR-26a/b, miR-30d, miR-106b-25, miR-29b, miR-30a, miR-133 and miR-590, and found that they participated in the process of AF [35]. Taken together, these findings show that miRNAs are involved in cardiac disorders and are differentially regulated in different cardiac diseases [36].

The let-7 family of miRNAs were the first human miRNAs to be discovered [37]. Many studies demonstrated that let-7 participates in various pathophysiological processes, such as cancer growth and formation [32], and the down- or up-regulation of certain let-7 family members has been observed in various types of tumor tissue [38]. Extensive evidence already suggests that let-7 functions as a tumor suppressor by targeting multiple oncogenes [38]. Among cardiovascular diseases, miR-7 was first reported to be associated with the risk of coronary artery disease [39]. Let-7e replacement was shown to ameliorate the abnormal up-regulation of beta adrenoceptor (b-AR) expression and markedly inhibit the incidence of arrhythmia in acute MI rats [32]. In addition, knocking down miR-7a/b has been shown to up-regulate collagen Ι expression and extend the fibrotic area and miR-7a/b overexpression has been shown to improve cardiac function, decrease fibrosis and exhibit an anti-fibrotic effect by targeting collagen I [39]. Therefore, in the present study, we investigated the specific miRNAs that modulate ECM protein (COL1A and COL3A) in hypoxic HL-1 cardiomyocytes. We found that miR-let-7 targeted COL1A and COL3A in both online miRNA databases and microarray data (Figure 1) and that this was down-regulated by hypoxia.

JNK activation has been observed in various cardiovascular diseases and it has been associated with a dramatic increase in AF propensity, including MI and heart failure [40]. A previous study demonstrated that stress-response kinase JNK in the atria was involved in arrhythmic remodeling by activating calcium/calmodulin-dependent protein kinase (CaMK) II and, in turn, up-regulating diastolic sarcoplasmic reticulum calcium leak, leading to aberrant intracellular waves and enhanced AF propensity [41]. Recently, we discovered the role of the JNK pathway in hypoxia-induced fibrosis and already confirmed that fibrotic protein under hypoxia can be expressed via the JNK pathway [6]. This research concluded that MAPK signaling could modulate fibrosis marker expression. Therefore, we hypothesize that miRNAs (miR-133a, miR-let-7a and miR-let-7e) may inhibit collagen indirectly through the JNK-ATF2 pathway induced by hypoxic stress.

There are several important findings in this study. A search through the Web miRNA database and microarray data identified miR-let-7 as targeting the fibrotic markers COL1A and COL3A. With various methods, including Western blotting, qRT-PCR and immunofluorescent assay, we proved the anti-fibrotic role of miR-133a, miR-let-7a and miR-let-7e on cardiomyocytes under hypoxic stress. Their corresponding miRNA mimic and antagomir modulation also influenced the fibrotic marker expression. Another novel finding is that an inhibition of JNK phosphorylation could enhance the expression of miRNAs and decrease the fibrotic collagen levels, which, in turn, might reduce the arrhythmogenic atrial remodeling. This study highlights the important role of miRNA in hypoxia-related atrial remodeling as well as atrial fibrosis (Figure 5).

There are some limitations of this study. The expression or protein level of collagens often varies during different progressing course of the underlying cardiovascular disease. Whether results derived from HL-1 cells can be readily extrapolated to atrial tissues for different cardiovascular diseases is uncertain. Although several molecular pathways may be implicated in atrial remodeling, fibrosis and the subsequent development of AF, the current study only focused on the role of miRNA on fibrotic markers and atrial fibrosis using hypoxic atrial HL-1 cells. Further investigations to determine the different molecular mechanisms and pathogenesis for AF associated with miRNAs are required. Finally, these miRNAs were only analyzed with cellular experiments and their effects in human heart tissue are not validated yet. Therefore, a future study with hypoxia induction and miRNA expression measurement in heart tissue will be conducted upon an approval from the institutional review board.

In conclusion, miRNA let-7a, -7e and -133a reduced atrial remodeling and fibrosis via the inhibition of the fibrotic markers COL1A or COL3A and post-transcriptional repression by the JNK pathway under hypoxic stress. This finding may provide new therapeutic strategies and future anti-arrhythmic drug development for atrial fibrillation.

## 4. Methods and Materials

### 4.1. Culture of Atrial Myocytes

The HL1 atrial cell line was derived from adult mouse atria and obtained from the National Taiwan University Hospital, Taipei (courtesy of Dr. Chia-Ti Tsai). The HL-1 cells were cultured in Claycomb medium [42] (Sigma) supplemented with 10% fetal bovine serum (FBS), 2 mM L-glutamine, 0.1 mM norepinephrine, 0.1 mg/mL streptomycin and 100 U/mL penicillin at 37 °C in a humid atmosphere of 5% CO_2_. The cells were plated at a density of 25,000 cells/cm^2^ on precoated plates (5 ng/mL fibronectin in 0.02% gelatin solution). Prior to the experiments, the cells were arrested overnight in reduced-serum media (2% FBS) [16].

### 4.2. Hypoxia Treatments and Material Preparation

For all assays, HL-1 cells were incubated in reduced-serum media (2% FBS). For hypoxia, the HL-1 cells were placed in a hypoxia chamber (NexBiOxy Inc., Hsinchu, Taiwan) and maintained at 37 °C with a humidified hypoxic atmosphere of 1% O_2_, 5% CO_2_ and 93% N_2_. Controls were maintained at 5% CO_2_ and 95% air at 37 °C. HL-1 was used as mirVana miRNA Mimic Negative Control #1 (100 nM, Applied Biosystems, Cat #4464058), mmu-miR-133a, mmu-let-7a mmu-let-7e (100 nM, Applied Biosystems, Cat #4464066, ID:MC12304, MC10050, MC12304) or inhibitor Negative Control, miRNA inhibitors (100 nM, GenePharma, Cat #B03001). The miRNA inhibitors were designed using the target mature miRNA and sequence as follows: mmu-miR-133a inhibitor, 5′-AAACCAGGGGAAGUUGGUCGAC-3′; mmu-let-7a inhibitor, 5′-ACUCCAUCAUCCAACAUAUCAA-3′; mmu-let-7e inhibitor, 5′-ACUCCAUCCUCCAACAUACCAA-3′; miRNA NC, 5′-UCUACUCUUUCUAGGAGGUUGUGA-3′. For certain experiments, cells were treated with inhibitors for 30 min before the hypoxic stimulus. JNK inhibition was conducted with 10 uM SP600125 inhibitor (purchased from Calbiochem, La Jolla, CA, USA) with a 10 mM stock solution in DMSO.

### 4.3. Western Blot Analysis

Total cellular protein was extracted from conditioned cells using Pro-prep protein extraction solution (iNtRON Biotechnology Inc., Seongnam-Si, Korea). The homogenates were centrifuged at 15,000× *g* for 20 min at 4 °C and stored at −20 °C. Nuclear fractions were obtained using a Kontes Dounce Homogenizer with hypotonic lysis buffer (10 mM HEPES, 1.5 mM MgCl_2_, 10 mM KCl, 0.5 mM DTT, 0.05% Igepal, pH 7.9). The lysate was centrifuged at 900× *g* for 5 min and then the nuclear pellet was lysed using Pro-prep protein extraction solution. Protein concentrations were determined using a DC Protein Assay (BioRad Laboratories, Inc., Hercules, CA, USA) and 25 μg of soluble protein per sample was electrophoresed on 8% or 12% SDS poly-acrylamide gels (SDS-PAGE) and then transferred to polyvinylidene difluoride (PVDF) membranes (Millipore).

After blocking with 3% bovine serum albumin (BSA) for 1 h at room temperature, membranes were incubated with an indicated antibody against HIF-1α (Santa Cruz Biotechnology, Novus Biologicals, Centennial, CO, USA), proCOL1A, proCOL3A (Santa Cruz Biotechnology), p-ATF2, p-MKK4, JNK/SAPK, pJNK/SAPK (pT183/pY185) (Cell Signaling Technology, Danvers, MA, USA), p38 or ERK (BD Biosciences, Bedford, MA, USA) at 4 °C overnight while a duplicate membrane was probed with anti-β-Actin (Geneway) as control. The membranes were then incubated with a species-specific horseradish-peroxidase-labeled secondary antibody (BD Biosciences: 1:5000) for 1 h at 37 °C. Between steps, membranes were thoroughly washed with TBST. Peroxidase activity was detected on an LA4000 system (Fuji) using ECL detection reagents (Millipore) and quantified using a luminescent imager (LAS-1000 Image Analyzer, Fujifilm, Berlin, Germany) and FluorChem imaging software (Alpha Innotech, San Leandro, CA, USA) [16].

### 4.4. MiRNA Microarray and Data Analysis

The miRNA microarray analysis was performed using Mouse & Rat miRNA OneArray^®^ v5 (Phalanx Biotech Group, Hsinchu, Taiwan), which contained triplicate 1265 unique miRNA probes from mice (miRBase Release 19.0) and 722 unique miRNA probes from rats (miRBase Release 19.0). It also contained 144 experimental control probes.

The assay started with a 2.5 μg total RNA sample using an miRNA ULSTM Labeling Kit (Kreatech Diagnostics, Amsterdam, The Netherlands). Hybridization was performed for 16 h on a microfluidic chip consisting of a chemically modified nucleotide coding segment complementary to the target miRNA (miRBase Release 19.0) or other control RNAs. Fluorescence images were captured by molecular devices, using an Axon 4000B scanner (Sunnyvale, CA, USA) and digitized using GenePix 4.1 software (Axon Instruments, Union City, NJ, USA). The signal intensity of each spot was processed with R software. Spotswith flag < B0 was filtered out and normalized with 75% media scaling normalization method. Normalized spot intensities were transformed to gene expression log2 ratios between the control and treatment groups. Data adjustments included data filtering, log2 transformation, gene centering and normalization. A two-sample t test was used for statistical analysis [43,44].

### 4.5. RNA Isolation and Quantitative Reverse Transcription-Polymerase Chain Reaction (qRT-PCR)

According to the manufacturer’s instructions, total RNA was extracted from HL-1 cells using a total RNA mini-kit (Geneaid, Taipei, Taiwan) and miRNAs were isolated and purified using a mirVana miRNA isolation kit (Ambion, Austin, TX, USA). RNA concentrations and A260/280 ratios were measured using a Thermo Scientific NanoDrop 2000 spectrophotometer. For mRNA analysis, total RNA was reverse transcribed to cDNA using TaqMan Reverse Transcription Reagents (Applied Biosystems, Branchburg, NJ, USA) and then subjected to qRT-PCR. Relative mRNA levels were ascertained by further amplification for the predicted genes with their respective primers using a Power SyBr Green Master Mix kit (Ambion). Glyceraldehyde 3-phosphate dehydrogenase (GAPDH) was used as the control to normalize the data. The levels of miRNAs were confirmed using mirVana qRT-PCR Primer Sets and a Step One real-time PCR system (Applied Biosystems, Foster City, CA, USA), according to the manufacturer’s instructions, with U6snRNA as a control. Tests were performed in duplicate and repeated three times.

### 4.6. Immunofluorescence

HL-1 cells were grown on gelatin-fibronectin-coated coverslips for 24 h, stimulated by hypoxia under various conditions, fixed with 4% paraformaldehyde in phosphate-buffered saline (PBS) for 15 min, permeabilized in 0.1% Triton-X100 for 10 min and then blocked with 1% BSA in PBST overnight at 4 °C. Afterwards, cells were incubated with antibodies against COL1A (ab 34710, Abcam) or COL3A (ab 7778, Abcam) (Santa Cruz Biotechnology) at 1:100 dilution in 1% BSA for 2 h at room temperature. After two washings with PBS, cells were incubated for an additional 60 min at room temperature with goat anti-mouse IgG conjugate or goat anti-rabbit IgG conjugate with fluorescein isothiocyanate (FITC) at 1:200 dilution in 1% BSA. Images were acquired with an IX-73 Inverted Microscope (Olympus, Tokyo, Japan) and counted using Cell Sense Dimension software (Olympus, Japan). HL-1 cells incubated exclusively with the secondary antibody were employed as a negative control. To measure staining intensity, digital images were sectioned into their basal and apical compartments and the densitometry of immunofluorescent staining was evaluated using the pixel intensity routine in ImageJ software (National Institutes of Health, Bethesda, MD, USA). Uniform microscope and laser settings were used for each experimental condition. Pixel intensity changes were expressed as percent increased or decreased, with further statistical analysis [16].

### 4.7. Luciferase Reporter Assay

To generate reporter vectors bearing miRNA-binding sites, fragments of the 3′-UTR of COL1A1 and COL3A1 (mouse) containing the exact target sites for miR-let-7 and miR-133a were generated. Briefly, COL1A1 or COL3A 3′-UTRs were inserted into multiple cloning sites at the downstream of luciferase gene (Aat II and EcoR I) in the pGL4.13 reporter vector (Promega, Madison, WI, USA). To generate COL1A and COL3A 3′-UTR containing miR-let-7- or miR-133a-mutated binding sites, site-directed mutagenesis was performed using wild-type 3′-UTR as the template. Luciferase activity was gauged at 48 h after transfection using a Synergy HT Microplate Reader (BioTek, Winooski, VT, USA). Firefly luciferase activity was normalized with beta-galactosidase activity to account for variations in the transfection efficiency among experiments [43].

### 4.8. Statistical Analysis

Quantitative data were expressed as mean ± SD of at least three independent experiments. Data were compared using the Student’s *t*-test for two groups and analysis of variance with post hoc *t*-test and corrections for *p* values was used for comparisons of more than two groups. The paired was used for comparisons of protein level before and after hypoxia or the antagonist. A *p* value < 0.05 (*) was considered to be statistically significant, which is shown in the respective figure captions [16].

## Figures and Tables

**Figure 1 ijms-23-09636-f001:**
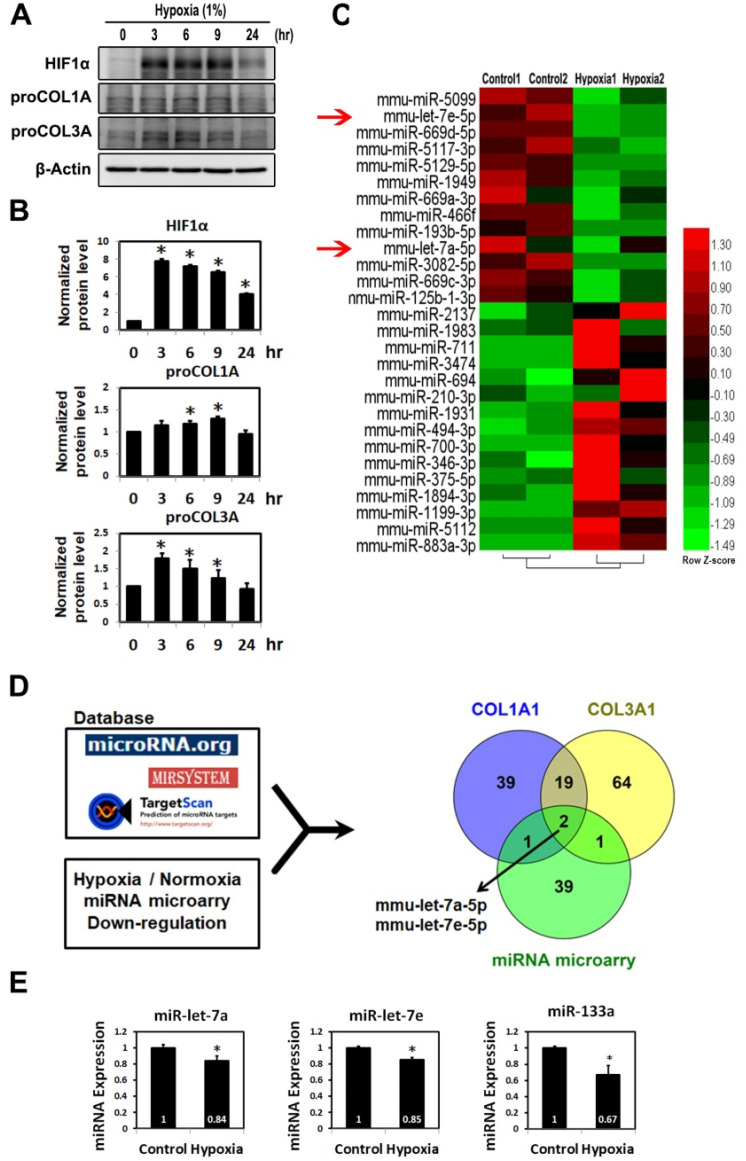
Differential expressions of proteins, mRNAs and microRNAs in hypoxic cardiomyocyte cells. (**A**) HL-1 cells were incubated under hypoxic conditions (1% O_2_) for the indicated times. The indicated proteins were detected in cellular homogenates by Western blot analysis. (**B**) qRT-PCR analysis of fibrotic mRNAs was performed. Error bars represent means ± SD from three independent experiments. * *p* < 0.01 vs. normoxia; vs. hypoxia + scramble control mimic (NC), *n ≥* 3. (**C**) Heat map showing the results of hierarchical clustering present the distinct microRNA expression profiles between hypoxia and controls. (**D**) Venn diagram showing the overlap between hypoxia-responsive microarray miRNAs or collagen-specific miRNAs in HL-1 cardiomyocytes under hypoxia stress. The three circles represent predicted miRNA target genes found in the list of reported miRNA websites. (**E**) qRT-PCR analysis of mmu-miR-133a, mmu-let-7a and mmu-let-7e miRNA expression in HL-1 cell. Mean ± SD, *n* ≥ 3, * *p* < 0.05.

**Figure 2 ijms-23-09636-f002:**
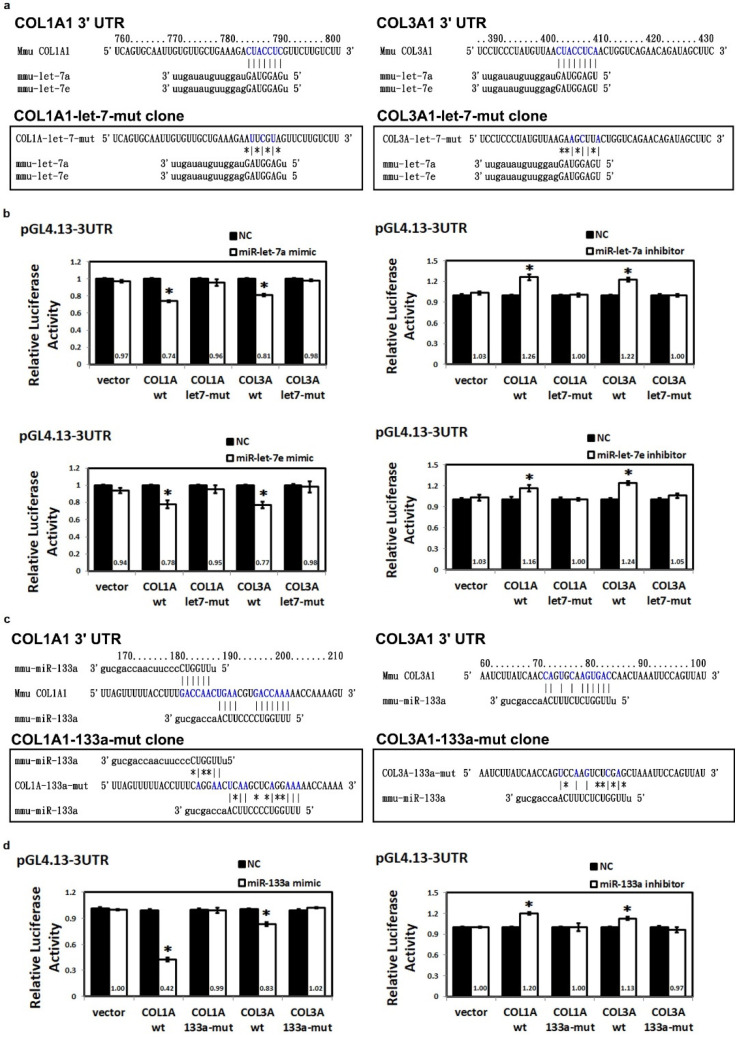
The miR133a and miR-let-7 family attenuated ECM markers through targeting collagen COL1A and COL3A. (**a**,**c**) The UCSC Genome Browser was used to align potential miRNA binding sites and mmu-miR-133a, mmu-let-7a and mmu-let-7e binding sites within the COL1A 3′ UTR and COL3A 3′ UTR. The COL1A 3′ UTR and COL3A 3′ UTR luciferase reporter constructs containing mutated target sites are shown with changes in the central nucleotides that should abolish binding of mmu-miR-133a or mmu-let-7 to the predicted target sites. miRNA binding sites were predicted using TargetScan and miRWalk. (**b**,**d**) Cells were transfected with a firefly luciferase and β-gal reporter construct, which contained 3′ UTR of COL1A or COL3A mRNA, along with either miRNA mimic, miRNA inhibitor or negative control (NC). For the mutant firefly luciferase reporter, putative miR-133a, miR-let-7a or miR-let-7e binding sites in 3′ UTR regions were also detected and normalized to beta-galactosidase reporter. Data are presented as mean ± standard error from at least three separate experiments. * *p* < 0.05 compared with the NCs. (COL1A = collagen type 1A1, COL3A = collagen type 3A1, UCSC = University of California, Santa Cruz).

**Figure 3 ijms-23-09636-f003:**
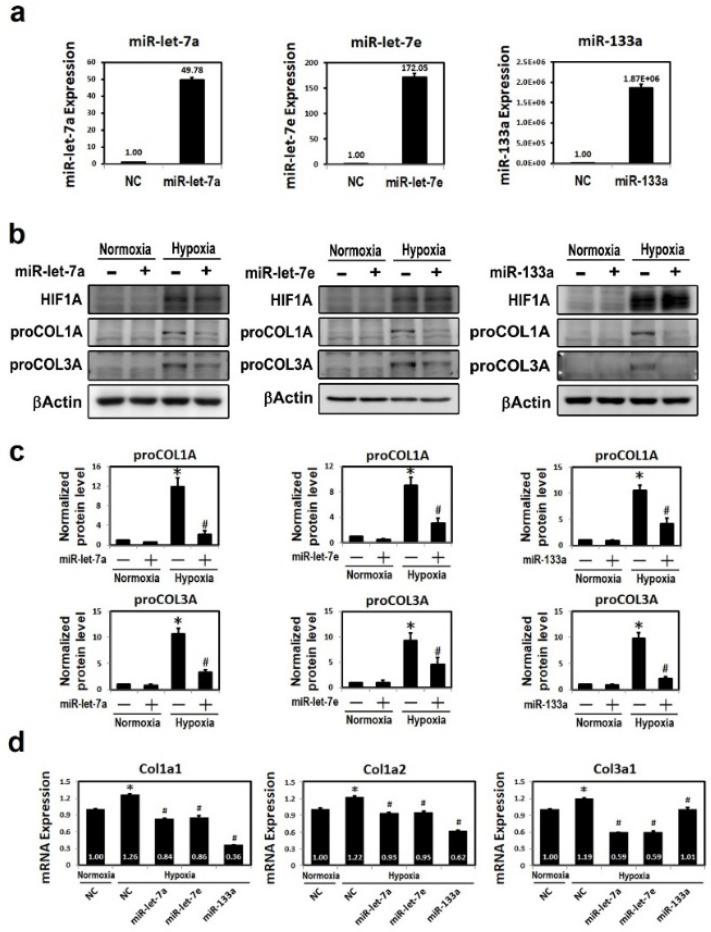
Regulation of collagen type 1A1 and 3A1 by miR-133a and let-7 family under hypoxic stress. (**a**) The HL-1 cells transfected with miRNA mimic were exposed to a hypoxic microenvironment and the levels of miR-133a, miR-let-7a and miR-let-7e were determined for transfection effectiveness from each group. (**b**) Western blotting analysis and quantification (**c**) of COL1A and COL3A protein expression in HL-1 cells transfected with the indicated mimic miRNA. (**d**) qRT-PCR analysis of COL1A1, COL1A2 and COL3A mRNA expressions in HL-1 cells transfected with the indicated mimic miRNA. Mean ± SD, *n* ≥ 3, * *p* < 0.05. (**e**) Effect of extracellular matrix shown by immunofluorescent green color expression in miR-133a and miR-let-7 family mimics treated in hypoxic HL-1 cells using an immunofluorescence assay. HL-1 cells were transfected with miR-133a mimic, miR-let-7a or miR-let-7e scramble control mimic (NC) and the expressions of COL1A and COL3A under hypoxic conditions were detected by immunostaining. Scale bar: 50 μm. Secondary antibody control slides for each group showed no COL1A or COL3A staining. The green intensity was quantified using ImageJ software and represents means ± standard error of the mean (SEM) from two independent experiments performed in duplicate (*n* > 150 cells, * *p* < 0.01 vs. normoxia, # *p* < 0.01 vs. hypoxia + NC control).

**Figure 4 ijms-23-09636-f004:**
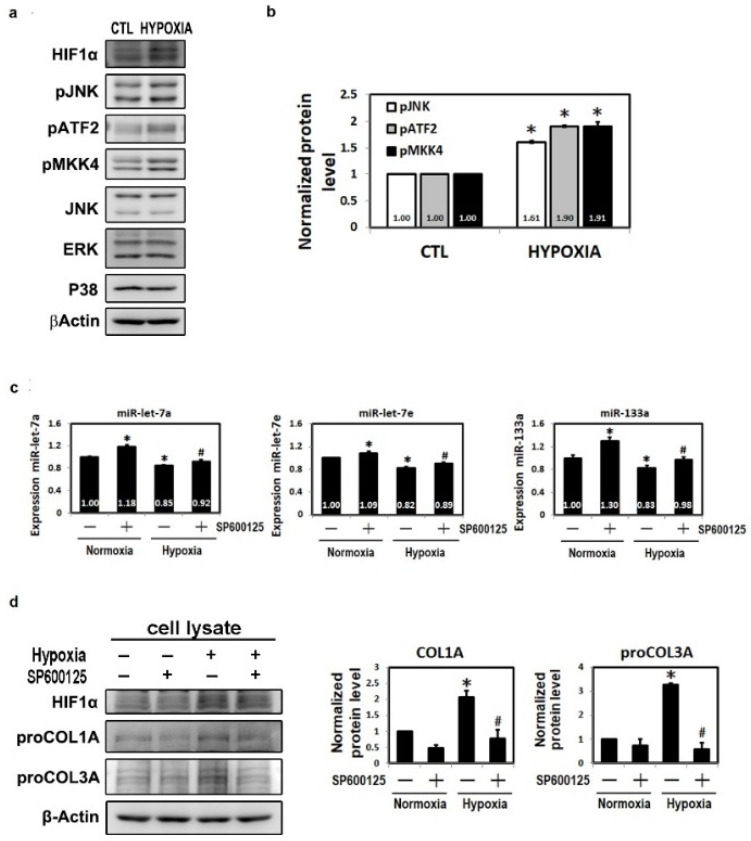
Post-translation repression of miR-let-7a, miR-let-7e and miR-133a by the JNK/SAPK pathway. (**a**) JNK pathway protein levels were evaluated by Western blotting and quantification (**b**) and HL-1 cells were treated with hypoxia. (**c**) Expressions of miR-133a, miR-let-7a and miR-let-7e were evaluated in HL-1 cells treated with SP600125 under hypoxic conditions. (**d**) Expressions of COL1A and COL3A proteins were analyzed by Western blotting and quantification. Mean ± SD, *n* ≥ 3, * *p* < 0.05 vs. normoxia; # *p* < 0.05 vs. hypoxia.

**Figure 5 ijms-23-09636-f005:**
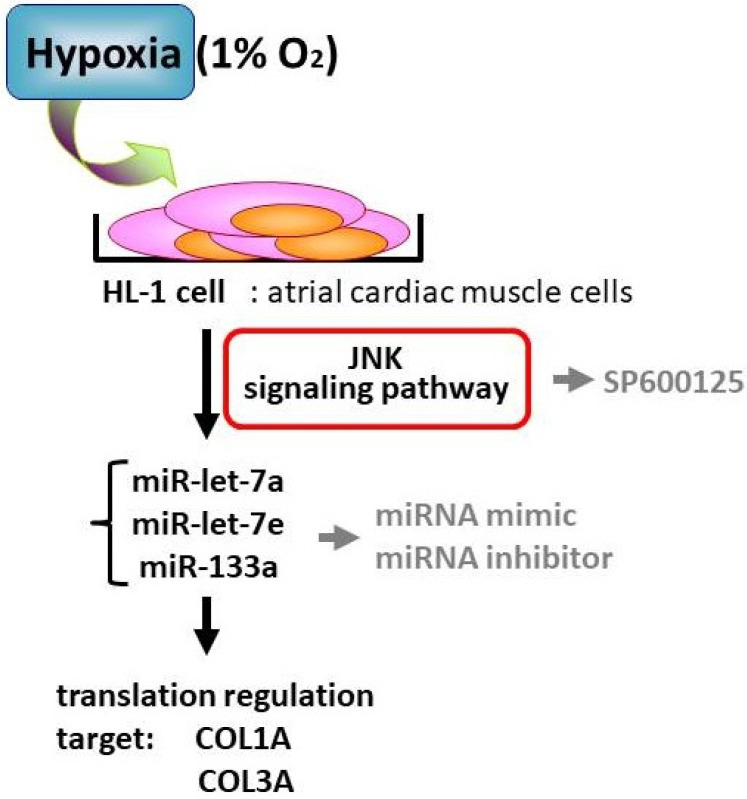
A proposed model of miRNA-mediated regulation of cardiac fibrosis under hypoxia by modulating JNK pathway.

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
