# Peer review of "MicroRNA Let-7a, -7e and -133a Attenuate Hypoxia-Induced Atrial Fibrosis via Targeting Collagen Expression and the JNK Pathway in HL1 Cardiomyocytes"

_ijms, 2022, doi:10.3390/ijms23179636_

Round 1

Reviewer 1 Report

1.     In Fig. 1c, what is the standard to determine miRNA expression value? In section 2.4, line 134-136, the authors claimed the value is determined based on “Normalized spot intensities were transformed to gene expression log2 ratios between the CONTROL and TREATMENT groups. Data adjustments included data filtering, log2 transformation, gene centering, and normalization.” However, the heat map in Fig. 1C showed that the expression levels of miR-5099 and let7-a-5p were increased in “control 1 and control 2” groups”. Moreover, specific miRNAs were decreased in both control and hypoxia groups. Is there another “CONTROL group” used for microarray normalization? 

Author Response

We answered the reviewer 1 comments which is attacked below. Thanks

Reviewer 2 Report

Study seems significant, however not novel. Authors need to clarify how their study is novel and contributes to the scientific literature.

Overall, manuscript has been drafted well, however, there are few following changes need to be made by the authors.

·         Aim of the study should be better cleared.

·         In line 34, there is a spelling mistake of indicible in hypoxia-indicible 34 factor (HIF), it is inducible. In line 43, previous research has been referred, authors need to clarify by proper citation, which research they are referring to.

·         Paragraphs in the introduction should be linked to make it coherent.

·         Results and conclusion should be improved for English language.

Author Response

We answered the reviewer 2 comments which is attacked below. Thanks

Round 2

Reviewer 1 Report

Although the authors provided the list of miRNA expressions in the authors’ response, the main concern regarding the heat map shown in Fig. 1C remains to exist. Please refer to following a previous study with a miRNA expression comparison between normoxia- and hypoxia-treated cells (https://www.ncbi.nlm.nih.gov/pmc/articles/PMC4174502/). Furthermore, the values shown in miRNA expression list do not match that shown in Fig. 1C and may lead to additional confusion.  

Author Response

Response:

Thank you for the valuable suggestion. According to reviewer’s suggestion, the Figure 1C has been revised and replaced according to following a previous study with a miRNA expression comparison between normoxia- and hypoxia-treated cells (PMC4174502). Moreover, the values shown in miRNA expression list are matching that shown in New Figure 1C. Thank you for the valuable suggestion, again. We hope that these changes and replies may meet your requirement for being published.

Round 3

Reviewer 1 Report

I have no further comments. The manuscript is acceptable.

This manuscript is a resubmission of an earlier submission. The following is a list of the peer review reports and author responses from that submission.

Round 1

Reviewer 1 Report

In this manuscript, the authors applied a cultured atrial cardiomyocyte cell line (HL-1) to examine the effect of hypoxia on regulating the expression of miRNA let-7a/e and miR-133a as well as atrial fibrosis-associated proteins, COL1A and COL3A. In addition to the usage of miRNA mimic and antagonist. Luciferase reporter assay with site mutation on 3’UTR was also used to determine the post-transcriptional regulatory relationship between miRNAs and collagens. Final a JNK inhibitor was used to determine the role of JNK activation in miRNA let-7a/e and miR-133a expression. Although the relationship among atrial fibrosis, miRNA, and collagen expression has been revealed in several separated reports, this study used a single HL-1 cell model to prove the linkage in a single study. Although this study touched on an important issue in the field of atrial fibrillation and fibrosis and provided certain meaningful information, this manuscript is not well-prepared. Besides, there remain several points needed to be clarified.     

1.     In an authors’ recent study, they used an identical cell model to examine the involvement of ROS-JNK signaling in hypoxia-induced atrial fibrosis in HL-1 cells (Int. J. Mol. Sci. 2021, 22(6), 3249). Of interest, the trends of HIF-1α, COL1A, and COL3A expression showed a different manner in the present and previous studies, particularly in the expression of HIF-1α at baseline and 24 hr-post hypoxia treatment, implying a variation in HIF-1α expression in their cultured system, such as that shown in Fig. 3b in the present study. However, in Fig. 1b, there is nearly no standard deviation in the statistic of HIF-1α expression. A re-check of quantitation and statistics is suggested.

2.     In Fig. 1c, what is the standard to determine miRNA expression value? According to the method shown in section 2.4. “Normalized spot intensities were transformed to gene expression log2 ratios between the control and treatment groups”, it is unreasonable that expression of let-7a was increased 8-fold in both control and hypoxia groups.    

3.     In Fig. 1e, what’s the time point for miRNA RT-PCR?

4.     Authors should explain why miRNA mimics not only downregulated protein expression of collagens but also decreased their mRNA expression. Is there an indirect regulation between miRNAs (let-7 and mir-133a) and collagen expression?

5.     What is “NT” in Fig. 3a? Does it mean “non-treatment”? Does Y-axis represent a fold increase of miRNA? According to the results of microarray and RT-PCR shown in Fig. 1, hypoxia did not dramatically reduce the expression of let-7 and mir-133a. Compared to other studies with transfection of miRNA mimics (such as that in Biochem Biophys Res Commun. 2012 Aug 31; 425(3): 668–672), the transfection efficiency of miRNA mimics in the present study is dramatically higher than that in other studies.. 

6.     Without fluorescence detection or quantitation, Fig. 3e cannot provide any meaningful information. If possible, in addition to RT-PCR, an in situ hybridization is suggested to validate the expression of miRNA mimics.   

Author Response

The file that response to Reviewer 1 Comments was attached 

Reviewer 2 Report

In their study with the title “MicroRNA let-7a, -7e and 133a attenuate hypoxia-induced atrial fibrosis via targeting collagen expression and the JNK pathway in HL1 cardiomyocytes” Lo et al. use the immortalised HL-1 mouse atrial cardiomyocyte cell line to identify differentially regulated microRNAs (miRNAs) under hypoxic conditions and investigate how these miRNAs affect atrial fibrosis by analysing collagen I and III expression.

The manuscript is generally well written (with a few exceptions in the use of English) and the authors perform control experiments that include expression of miRNA antagonists or mimetics. However, as the authors also recognise, the use of this cell line is not suitable on its own for drawing conclusions on atrial fibrosis. The work and its conclusions could be re-enforced if experiments were also conducted at the organ level or even with human induced-pluripotent stem cell-derived atrial cardiac myocytes.

 Furthermore, the authors use two protocols for hypoxia (hypoxic chamber or cobalt chloride) but, apart from the experiment in Figure 1a and 1b, it is not clear which one is used each time and whether both protocols produce similar results.

Other comments are the following:

  • In the Methods, the title in 2.1 also mentions culture of atrial fibroblasts. However, there is no description of culture or results from such cells in the manuscript.
  • Line 48 (Introduction): the authors wrongly cite reference 5 as their own work.
  • A description of the transfection of miRNAs antagonists or mimetics is missing.
  • Immunofluorescence: By eye and taking into account the cell size, there seems to be no difference in pro-collagen expression between the different conditions. The description also of Figure 3e in the respective legend differs from the description in the results section. Is it pro-collagen 1 and 3 or collagen 1 and 3 detected by immunofluorescence?
  • The authors do not provide any information about the final concentration of the MAPK inhibitors used and this precludes reproduction of the results by others.

Author Response

The file that response to Reviewer 2 comments and figures was attached.

Round 2

Reviewer 2 Report

This is an evaluation of the revised manuscript ““MicroRNA let-7a, -7e and 133a attenuate hypoxia-induced atrial fibrosis via targeting collagen expression and the JNK pathway in HL1 cardiomyocytes”. The reviewer maintains the opinion that at least some experiments need to be performed at the organ (heart) level to show the relevance with the results in the HL1 cells.

Comments:

1.       Line 215: The authors cannot claim that they investigate myocardial fibrosis when they do not use heart tissue. The authors should perform hypoxia/ischemia in hearts and measure miRNA expression (at least for the miRNAs presented in this study) in heart tissue. Do they then observe similar results as that in HL1 cells?

2.       The English language in the corrections/additions and in some other parts of the revised manuscript needs improvement.

3.       The authors have deleted reference 5 but the numbering of the references has not changed compared to the previous version of the manuscript.

4.       Even if the authors deleted reference 5, it still appears at line 51 in page 2. The authors need to provide at that point the correct reference of their own previous work.

5.       The authors should provide catalogue numbers for each miRNA mimic, inhibitor or negative control they have used, if these miRNA reagents where readily available from the companies they were purchased. This will enable replication of the study by others.

6.       Please provide catalogue numbers for all antibodies used. Especially, for the immunofluorescence experiments with COL3A they should mention that two antibodies from two different companies were used.

7.       The authors should replace Figure 3e with SubFig3 or at least add the quantification graphs to the main Figure. They should also check their graphs for spelling errors.